**Data Availability Statement:** The data is available from the Ethiopian Public Health Institute data management center at https://ndmc.ephi.gov.et.

# Prevalence and factors associated with hepatitis B and C virus infections among female Sex workers in Ethiopia: Results of the national biobehavioral Survey, 2020

Birra Bejiga Bedassa[1]◎*, Gemechu Gudeta Ebo[1]◎, Jemal Ayalew Yimam[1,2]◎, Jaleta Bulti Tura[1‡], Feyiso Bati Wariso[1‡], Sileshi Lulseged[3‡], Getachew Tollera Eticha[1‡], Tsigereda Kifle Wolde[1‡], Saro Abdella Abrahim[1]◎

**1** Ethiopian Public Health Institute, Addis Ababa, Ethiopia, **2** Department of Statistics, College of Natural Science, Wollo University, Kombolcha, Ethiopia, **3** Faculty of Medicine, College of Health Sciences, Addis Ababa University, Addis Ababa, Ethiopia

◎ These authors contributed equally to this work.
‡ JBT, FBW, SL, GTE and TKW are contributed less but substantially.
* birr4allephi@gmail.com

## Abstract

### Background

Hepatitis B and C virus infections are endemic diseases in sub-Saharan Africa, the region with the highest prevalence of these infections in the world. Female sex workers are exposed to sexually transmitted infections, including hepatitis B and C, because of their high-risk sexual behavior and limited access to health services. There are no large-scale data on the prevalence of hepatitis B and C virus infections among female sex workers in Ethiopia, a critical gap in information this study aimed to fill.

### Methods

**This was a** cross-sectional, biobehavioral survey conducted from December 2019—April 2020 among 6085 female sex workers aged ≥15 years and residing in sixteen (16) regional capital cities and selected major towns of Ethiopia. Blood samples were collected from the participants for hepatitis B and C virus serological testing. The data were collected using an open data kits (ODK) software and imported into STATA version16 for analysis. Descriptive statistics (frequencies and proportions) were used to summarize data on the study variables. Bivariable and multivariable logistic regression analyses were conducted to determine the strength of association between independent variables (risk factors) and the outcome (hepatitis B and C virus infection). Adjusted Odd ratio (AOR) was used to determine independent associations, 95% confidence interval to assess precision of the estimates, and a *P* value ≤ 0.05 to determine statistically significant.

### Results

The prevalence of hepatitis B and C infections among the 6085 female sex workers was 2.6% [(95% CI (2.2,2.8)] and 0.5% [(95% CI (0.4,0.7)], respectively. Female sex workers

following a request sent to Mr. Lemessa Negeri the EPHI Data Manager, at the e-mail address (nlemessa@gmail.com) or at Fax:011 275 8634. P. O. Box: 1242 or 5654 of Ethiopian Public Health Institute, Addis Ababa, Ethiopia. The authors of this manuscript do not have any special privilege in accessing the data.

**Funding:** This project was conducted using U.S. President's Emergency Plan for AIDS Relief (PEPFAR) funds obtained through the U.S Centers for Disease Control and Prevention (CDC) under Cooperative Agreement #U2GGH001226 and the Government of Ethiopia Federal HIV Prevention and Control Office (FHAPCO) budget line ETH-H-HAPCO-1553. The findings and conclusions in this report are those of the authors and do not necessarily represent the official position of the funding organizations. The funders had no role in study design, data collection, analysis, decision to publish, or preparation of the manuscript.

**Competing interests:** The authors have declared that no competing interests exist.

who had 61–90 and $\geq$91 paying clients in the past six months [(AOR = 1.66; 95% CI, (0.99, 2.79); $P$ = 0.054] and [(AOR = 1.66 95% CI, (1.11, 2.49); $P$ = 0.013], respectively, age at first sex selling of 20–24 and >25 years [(AOR = 1.67; 95% CI, (1.14, 2.44); $P$ = 0.009)] and [(AOR = 1.56; 95% CI (1.004, 2.43); $P$ = 0.048)], respectively, known HIV positive status [(AOR = 1.64; 95% CI (1.03, 2.62); $P$ = 0.036] were significantly associated with the prevalence of hepatitis B virus infection. Similarly, hepatitis C was significantly associated with, age at first sex $\leq$15 years and age 16–20 years [(AOR = 0.21; 95%CI (0.07,0.61); $P$ = 0.005)] and [(AOR = 0.18; 95% CI (0.061, 0.53); $P$ = 0.002)], respectively, known HIV positive status [(AOR = 2.85; 95%CI (1.10,7.37); $P$ = 0.031)] and testing positive for syphilis [(AOR = 4.38; 95% CI (1.73,11.11); $P$ = 0.002)], respectively.

## Conclusion

This analysis reveals an intermediate prevalence of hepatitis B and a low prevalence of hepatitis C infection among female sex workers in Ethiopia. It also suggests that population groups like female sex workers are highly vulnerable to hepatitis B, hepatitis C, and other sexually transmitted infections. There is a need for strengthening treatment and prevention interventions, including immunization services for hepatitis B vaccination, increasing HCV testing, and provision of treatment services.

## Introduction

Viral hepatitis is an important public health problem globally, and hepatitis B virus (HBV) and hepatitis C virus (HCV), in particular, are endemic in developing countries [1]. It is estimated that 325 million people were living with HBV and HCV in 2019 globally, and hepatitis was responsible for an estimated 1.4 million deaths per year mostly from hepatitis-related cirrhosis and liver cancer [2]. Unlike most communicable diseases such as tuberculosis (TB), Human Immunodeficiency Virus(HIV), malaria and diarrhea that declined sharply, the absolute burden and relative rank of viral hepatitis increased between 1990 and 2013 [3, 4]. According to the 2017 Global Hepatitis Report, viral hepatitis caused 1.34 million deaths in 2015, a death toll that was higher than that caused by HIV infection and comparable with that of tuberculosis [5].

In the African Region, HBV infection is highly endemic and affects an estimated 5%-8% of the population, mainly in West and Central Africa [6]. People living with HIV (PLHIV) are at high risk of becoming ill and dying from hepatitis. Some 2.6 million PLHIV are co-infected with HBV, and some 2.3 million with HCV [7]. The World Health Organization (WHO) 2019 Progress Report on HIV, viral hepatitis, and sexually transmitted infections (STI), showed, in the sub-Saharan Africa (SSA), there were over 60 million cases of chronic hepatitis B and over 10 million cases of chronic hepatitis C infections [8].

In SSA, female sex workers (FSW) had a high-risk behavior and remain important in terms of transmission of HBV, HCV, HIV/AIDS and other STI acquisition and transmission. This could be because FSW have numerous sex partners and they were engaged in unprotected and other forms of sex that cause contact with body fluids of a partner who has STI. FSW are often in a weaker position to negotiate safe sex because of social, economic, cultural and legal reasons [9].

Female sex work is a high-risk activity associated with HBV, HCV, and several other STI [7, 10]. The higher risk of getting infected with HIV and other STI, such as syphilis and hepatitis among FSW is primarily associated with the high number of sexual partners and increased

frequency of unprotected sex [11, 12]. Several studies have shown that low adherence to condom use, multiple sexual partners, unsafe sexual practices, illicit drug use, and co-infection with other STI increase the risk of HBV and HCV transmission [13, 14]. FSW also have a higher risk of contracting STI from their non-paying partners than from their paying clients [13, 14].

In Ethiopia, FSW carry a disproportionate burden of HBV, HCV, and HIV infection [15]. According to the Ethiopian Demographic and Health Survey (EDHS) 2016 report, the marked regional variation that was driven by most at-risk populations (MARPS) indicates that urban areas and females are more affected than rural areas and males, respectively [16]. Small towns are also becoming hotspots and can potentially bridge further the spread of the HIV and HBV infections to rural settings, where the female are twice more affected than males [16].

Ethiopia is in the region where HBV prevalence among the general population is considered hyper-endemic with a prevalence of 8%-12%, and that of HCV prevalence estimated at not less than 2.5% [17]. An earlier study conducted in Ethiopia reported that 12% of hospital admissions and 31% of the mortality on the medical wards in Ethiopian hospitals were due to chronic liver disease (CLD) [17]. However, there isn't much done and the available data on associated chronic liver disease or hepatocellular carcinoma are not sufficient.

There were no data indicating a national prevalence of HCV among FSWS, however the report from Dessie City, Northeast Ethiopia and Gondar Town, Northwest Ethiopia indicates the prevalence of HCV among FSWs 0.6% and 6.7% respectively [12]. The lack of access to regular screening of STIs among FSWs contributes a lot to the disease's transmission in resource-limited countries, including Ethiopia and the national study on HCV will be a pre requisite for this study groups [18].

There are limited data from isolated studies among FSW showing the prevalence of HBV infection in Ethiopia Hawassa 9.2% [19], Gonder 28.9% [20], Mekelle 6% [21], and Dessie 13.1% [12]. FSW have been identified as a population group with the highest risk for STI, including HBV and HCV, and should perceive priority in the national HIV/AIDS program [22]. Although there is an ongoing HBV, HCV, and other STI program in the country, there is no national data among FSW to determine HBV and HCV prevalence and driving factors. Therefore, the current study was conducted to explore the prevalence of HBV and HCV infections and identify the factors associated with these infections among FSW in Ethiopia.

## Materials and methods

### Study setting and population

The study was done in Ethiopia, a country divided into eleven regions and two city administrations with a total population of 120 million and had low per capita income [15]. In 2016, when the country started a national viral hepatitis prevention and control program [23]. Regional capital cities and selected towns with the highest number FSW (hotspots), including Adama, Addis Ababa, Arba Minch, Bahir Dar, Combolcha/Dessie, Dilla, Dire Dawa, Gambella, Gonder, Harar, Hawassa, Jimma, Logia/Semera, Mizan, Nekemite, and Shashemane were involved in the study. The cities and towns constituting the study sites were selected purposively, and included the capital cities of all regions of and major towns of the country. These are home to large numbers of government and nongovernment employees, migrant workers, and secondary, college, and university students as well as transient populations that temporarily reside in these cities and towns including long-distance trucks drivers, migrant workers, and businessmen.

### Study design and period

This was a cross-sectional, nation-wide, biobehavioral study conducted among FSW aged ≥15 years during the period from December 2019—April 2020.

## Target population

The target population of the study is all FSW living in cities and towns in Ethiopia, which constitute the hotspots of female sex work in the country.

## Study population

FSW aged ≥15 years residing in regional capitals and selected major towns who worked in these cities and towns in the last one month preceding the survey. The survey included both fixed (venue-based) and floating (street-based) FSW.

## Inclusion and exclusion criteria

We included women aged ≥15 years, who received money/other benefits in exchange for sex with four or more people within the last 30 days, agree to participate in the survey including interviewing and biological testing, able to provide informed consent and communicate in one of the survey languages, had a valid coupon provided by the study team, and residing or working in the survey city or town for the last one month. FSW residing outside the selected survey sites and those who did not provide consent of the interview and blood sampling were excluded from the survey.

## Sample size and sampling procedure

The sample size was determined based on a single population proportion formula using the following inputs: 95% confidence interval, α = 0.05, margin of error (d) of 35%, and proportion (p) of 2% [23], and a design effect (DEFF) with a replacement for non-responders. With these assumptions, the minimum desired sample size of FSW in sixteen (n = 16) major regional capitals and selected major towns with FSW hotspots was 6085 after adding 10% contingency. This was divided and assigned to 16 sites proportionate to population size.

Specific hotspot areas for FSW were identified during tools and procedures pretesting with support from HIV/AIDS Prevention and Control office (HAPCO), woreda (district) health offices, and drop-in-clinics (DICs), local organizations working with FSW. We used a respondent-driven consecutive sampling using a standardized questionnaire for recruitment of study participants. The local organizations assisted in identifying the initial respondents of the survey, referred to as "seeds". The number of seeds for each site was determined based on the result of a formative assessment. Five "seeds" for each site with allocated sample of <450, six-eight seeds for each site with sample 450–900, and 12 seeds for each site with sample of 1101 were recruited. The "seeds" were selected based on the type of sex worker, age category, and geographic location of the site. These include those FSW who were bar- and/or hotel-based, red lighthouses, local drinking houses, street-based and hidden (cell phone-based).

FSW with a known social network were given each three coupons for use to invite her friends or other FSW contacts who were in her network. This approach helped in reaching as many eligible FSW as possible. The coupon remained active from the day it was given to the potential participant and expired after two weeks or if the study was completed earlier. We used anonymous fingerprint-based code obtained using biometric fingerprint scanners to ensure that all respondents participated only once. This was not linked to the biobehavioral questionnaire and was used only for avoiding multiple enrollments.

Damaged, mutilated, not readable, photocopied, not sealed/stamped coupon was considered not valid. Each participant coming to the study site would need to bring her coupon that was identified by specific number given by the referring person. New participants were given coupons and asked to recruit three additional acquaintances. This process continued until the

desired sample size was achieved and respondent-driven sampling (RDS) equilibrium condition attained. Progress towards reaching the equilibrium was monitored by using key parameters including current HIV status, type of sex work, and consistent condom use.

## Data collection procedure and data quality management

Data were collected using a pre-tested structured questionnaire initially developed in English and then translated into the local language (Amharic) was entered onto open data kits (ODK) software. Training was provided to the study team, coordinators, interviewers, blood sample collectors, coupon managers (for RDS), receptionists (for RDS) and accompanying referral liaisons. The training included different topics with a focus on the survey sampling methodology, procedures, and data collection tools, and overall study site management. Data collection tools and questionnaires were pretested in a pilot survey in Bishoftu town, a site not included in the main study. Feedback from the pilot was used to finalize the data collection tools, capturing process, field logistics, and operational procedures.

## Sample collection and processing

The survey used whole blood for the rapid HIV, hepatitis-B and syphilis testing. After collecting whole blood of 5 ml using EDTA tube; HIV testing, Hepatitis B surface antigen (HBsAg), hepatitis C antibody (HCVAb) and syphilis testing were performed right after sample collection. Then after centrifuging the whole blood, the plasma was separated and aliquoted in two a 1.8 ml preprinted labeled nunc tube for viral load quantification and quality control testing. The plasma was separated and stored using two nunc tubes vials (1.8 ml) under -20 ˚C in a nearby health institution/regional laboratory until collected by survey teams from Ethiopian Public Health institute (EPHI). The stored sample were transported to EPHI on a weekly basis enclosing sample transport forms (tracking sheet indicating the samples details) by the sample collectors (laboratory personnel). Plasma specimens were transported by using triple-packaged Iceboxes (with Ice packs, less than 0˚C). Until testing, the plasma specimens were stored at -80˚C at EPHI National HIV Reference Laboratory. After viral load testing the remaining plasma samples were stored for future further testing. Once a participant's positive, linkage to service was made.

## Linkage to care and treatment

We had collaboration with FSW outreach programs, the associations of people living with HIV and local key population-friendly HIV/STI clinics for linkage to care and treatment services. Referral forms were completed for the survey participants to facilitate linkage to the service facility in the participant residential city or town. Health care providers delivered the care and treatment to all referred FSW as required.

**Biological analysis.** HBsAg and HCVAb were screened by using a rapid test kit according to manufacturer principles and procedure. Initially, the VIRUCHECK Rapid One Step HBsAg Test kit was utilized to determine HBV infection status and Flaviscreen Plus HCV Diagnostic Test Kit, was used to determine the infection status of HCV Ab. Ethiopian Food, Medicine, and Health Care Administration and Authority (FMHACA) had approved this kit for use in Ethiopia. Positive samples were sent to the Ethiopian Public Health Institute (EPHI) and confirmed by using an enzyme-linked immunosorbent assay (ELISA). Syphilis was screened using Chembio Dual Path Platform (DPP) Syphilis Screen and Confirm Assay, according to manufacturer principles and procedures in the kit insert. Testers were trained on how to use the test kit and control lines. All invalid results were repeated. Quality control (QC) panels consisting of a positive and negative control specimen were done in parallel with the testing procedure to

ensure test kits were performing correctly. Other STI, including HIV and syphilis, were tested using the national rapid testing algorithm and the results were returned during the second study visit.

## Data analysis

The data was collected using the ODK software on tablet computers, and was exported to MS-EXCEL, cleaned, and imported to STATA Verssion16 for analysis. The RDS recruitment process (Tree of recruitment), assessment of the RDS assumptions, and RDS weight generating were implemented using the RDS package inbuilt in R statistical software [24]. Homophily and convergence, the common assumptions in RDS, were checked in HIV status, consistent condom use, and type of FSW and met the RDS criteria. The RDS weights were exported using the RDS-II function to STATA and merged with the whole dataset for further analysis. Descriptive statistics like the crude and RDS adjusted frequency, mean and standard deviation were calculated. Bivariate and multivariate logistic regression analyses were conducted to determine the strength of association between independent variables (risk factors) and the outcome variables (hepatitis B and C virus infection). Variables achieving a *P* value of <0.2 in the bivariate analysis were included in the multiple logistic regression model. Strength of association was measured using Adjusted Odd Ratios (AOR), precision of estimates determined using 95% confidence intervals, and a *P* value ≤ 0.05 was used as cut-off to determine statistical significance.

## Ethical consideration

Ethical approval for the study protocol was obtained from the Scientific and Ethical Research Office (SERO) of the Ethiopian Public Health Institute (EPHI). Potential participants were told about the study purpose and procedures, potential risks, and protections using the local language. Written informed consent was obtained from each survey participant for the interview, blood sample collection, and storage of biospecimens for future testing. Participants were compensated for their time and reimbursed the transport costs.

## Results

### Socio demographic characteristics

A total of 6085 FSW participated in the survey. Their median age [Interquartile Range (IQR)] was 25 (8) years, and the highest number, 1980 (32.5%), were in age group 20–24 years and the lowest 615 (10.1%) in the age group 15–19 years (Table 1). A majority, 5031(82.7%), of them had formal education, 2946 (48.4%) were never married, and 4212 (69.2%) were pregnant a least once. A majority, 5694 (93%), of the respondents reported that selling sex was their main source of income, and 2066 (34%) of them had an average monthly income of Ethiopian Birr (ETB) 2500–4999 equivalent of U.S Dollar (USD) 60–120.

### Distributions of FSW by study cities/towns

A majority, 1101 (18.11%), of the 6085 FSW resided in Addis Ababa, the national capital followed by Adama, 676 (11.1%). A smaller proportion of them were in other cities/towns (Table 2).

### Sexual and behavioral characteristics

The median (IQR) age at first sex was 16 (3) years. The majority, 3384 (55.6%), of respondents had the first sex between the age of 16 and 20 years (Table 3). Some 2335 (38%) of the

**Table 1. Socio-demographic characteristics among female sex workers in cities/towns, Ethiopia, 2020 (N = 6085).**

| Variables | | Frequency | Percent | 95% CI |
|---|---|---|---|---|
| Age (years): Median (IQR) = 25 (8). | 15–19 | 615 | 10.1 | 9.4–10.9 |
| | 20–24 | 1980 | 32.5 | 31.4–33.7 |
| | 25–29 | 1815 | 29.8 | 28.7–31.0 |
| | 30–34 | 856 | 14.1 | 13.2–15.0 |
| | 35–59 | 819 | 13.5 | 12.6–14.3 |
| Level of education | Non-formal Education | 1054 | 17.3 | 16.4–18.3 |
| | Primary 1st cycle (grade 1–4) | 848 | 13.9 | 13.1–14.8 |
| | Primary 2nd cycle (grade 5–8) | 2712 | 44.6 | 43.3–45.8 |
| | Secondary school and above | 1471 | 24.2 | 23.1–25.3 |
| Marital status | Married/Cohabitation | 231 | 3.8 | 3.3–4.3 |
| | Divorced/Separated/Widowed | 2908 | 47.8 | 46.5–49.0 |
| | Never married | 2946 | 48.4 | 47.2–49.7 |
| Ever been pregnant | No | 1873 | 30.8 | 29.6–31.9 |
| | Yes | 4212 | 69.2 | 68.1–70.4 |
| Number of pregnancies | 0 | 1873 | 30.8 | 29.6–31.9 |
| | 1 | 2059 | 33.8 | 32.7–35.0 |
| | 2 | 1238 | 20.3 | 19.3–21.4 |
| | 3+ | 915 | 15 | 14.2–16.0 |
| Currently pregnant | No | 5973 | 98.2 | 97.8–98.5 |
| | Yes | 112 | 1.8 | 1.5–2.2 |
| The main source of income | Other than sex work | 391 | 6.4 | 5.8 _ 7.1 |
| | Sex work | 5694 | 93.6 | 92.9–94.2 |
| Average monthly income from selling sex (ETB) | < 2500 | 1778 | 29.2 | 28.1–30.4 |
| | 2500–4999 | 2066 | 34 | 32.8–35.1 |
| | 5000–7499 | 1175 | 19.3 | 18.3–20.3 |
| | 7500+ | 1066 | 17.5 | 16.6–18.5 |

**Table 2. Distributions of female sex workers by cities/town in Ethiopia, 2020 (N = 6085).**

| Residence (City/Town) | Frequency | Percent | 95% CI |
|---|---|---|---|
| Adama | 676 | 11.1 | 10.3–11.9 |
| Addis Ababa | 1101 | 18.1 | 17.1–19.1 |
| Hawassa | 522 | 8.6 | 7.9–9.3 |
| Gambella | 468 | 7.7 | 7.0–8.4 |
| Diredawa | 434 | 7.1 | 6.5–7.8 |
| Bahir Dar | 372 | 6.1 | 5.5–6.7 |
| Jimma | 254 | 4.2 | 3.7–4.7 |
| Mizan | 255 | 4.2 | 3.7–4.7 |
| Nekemite | 257 | 4.2 | 3.7–4.8 |
| Arba Minch | 251 | 4.1 | 3.6–4.6 |
| Combolcha/Dessie | 251 | 4.1 | 3.6–4.6 |
| Dilla | 251 | 4.1 | 3.6–4.6 |
| Gonder | 250 | 4.1 | 7.0–8.4 |
| Logia/Semera | 251 | 4.1 | 3.6–4.6 |
| Shashemane | 250 | 4.1 | 3.6–4.6 |
| Harar | 242 | 4.0 | 3.5–4.5 |

**Table 3. Sexual and behavioral characteristics of female sex workers in cities /towns, Ethiopia, 2020 (N = 6085).**

| Variables | | Frequency | Percent | 95% CI |
|---|---|---|---|---|
| Age at first sexual intercourse (years), median (IQR) | 16 (3) | | | |
| Age at first sexual intercourse (years), n (%) | ≤15 | 2430 | 39.9 | 38.7–41.2 |
| | 16–20 | 3384 | 55.6 | 54.4–56.9 |
| | ≥21 | 271 | 4.5 | 4.0–5.0 |
| Age at first sex selling | < 20 | 2328 | 38.3 | 37.1–39.5 |
| | 20–24 | 2348 | 38.6 | 37.4–39.8) |
| | ≥25 | 1406 | 23.1 | 22.1–24.2 |
| Age of first sex partner | 5 or more years younger | 73 | 1 | 0.8–1.6) |
| | About the same age | 2335 | 38.4 | 37.2–39.6 |
| | 5–10 years older | 2425 | 39.9 | 38.6–41.1 |
| | More than 10 years older | 1252 | 20.6 | 19.6–21.6 |
| Location of sexual practice | Hotel/bar-based | 2023 | 33.2 | 32.1–34.4 |
| | Street-based | 1871 | 30.7 | 29.6–32.0 |
| | Home-based | 348 | 5.7 | 5.2–6.3 |
| | Any type** | 661 | 10.8 | 9.4–11.2 |
| First sex experience | Wanted | 4738 | 77.9 | 76.8–78.9 |
| | Forced | 1347 | 22.1 | 21.1,23.2 |
| Number of paying partners in the last 6 months, n (%) | ≤ 30 | 2308 | 37.9 | 36.7–39.2 |
| | 31–60 | 1436 | 23.6 | 22.5–24.7 |
| | 61–90 | 696 | 11.4 | 10.7–12.3 |
| | ≥91 | 1645 | 27 | 25.9–28.2 |
| Ever had Anal intercourse, n (%) | Yes | 426 | 7 | 6.4–7.7 |
| | Never | 5659 | 93 | 92.3–93.6 |
| Used condom during anal sex (n = 426) | Yes | 252 | 59.2 | 54.4–63.7 |
| | No | 174 | 40.8 | 36.3–45.6 |
| Consistent Condom utilization during the last 30 days with paying clients | Yes | 5119 | 84.1 | 83.2–85.0 |
| | No | 966 | 15.9 | 15.0–16.8 |
| HBV among non-condom users during the last 30 days with paying clients | HBsAg positive | 23 | 2.4 | 1.6–3.5 |
| | HBsAg negative | 943 | 97.6 | 96.5–98.4 |
| HBV among condom users during the last 30 days with paying clients | HBsAg positive | 134 | 2.7 | 2.2–3.2 |
| | HBsAg negative | 4651 | 97.3 | 96.9–97.8 |
| Breakage of condom during the last 30 days | No breakage | 4260 | 70 | 68.8–71.1 |
| | Experienced breakage | 1825 | 30 | 27.4–31.3 |
| In the last 30 days, used cigarettes or cigars | No | 5343 | 87.8 | 87.0–88.6 |
| | Yes | 742 | 12.2 | 11.4–13.0 |
| Alcohol consumption level (AUDIT scores) | Not Risky | 2594 | 42.8 | 41.6–44.0 |
| | Harmful, hazardous drinking | 1210 | 20 | 19.0–21.0 |
| | Alcohol dependence indication | 2257 | 37.2 | 36.0–38.5 |
| Chewing khat in the last 30 days | Yes | 3827 | 62.9 | 61.7–64.1 |
| | No | 2258 | 37.1 | 35.9–38.3 |
| Any drug used other than alcohol and khat in the last 30 days | Yes | 700 | 11.5 | 10.7–12.3 |
| | No | 5382 | 88.4 | 87.7–89.3 |
| Used shisha in the last 30 days | Yes | 856 | 14.1 | 13.2–15.0 |
| | No | 5229 | 85.9 | 85.0–86.8 |

(*Continued*)

**Table 3.** (Continued)

| Variables | | Frequency | Percent | 95% CI |
|---|---|---|---|---|
| Depression level | Not depressed | 2468 | 40.6 | 39.3–41.8 |
| | Mild depression | 2525 | 41.5 | 40.3–42.7 |
| | Moderate to severe depression | 1092 | 17.9 | 17.0–18.9 |

*Alcohol Use Disorders Identification Test (long version)

**Any type: Restaurant/cafe /cake bet, Local drink house (arake bet, tella bet, tej bet), SPA/massage/beauty, Redlight

respondents had first sex with a person of their age, while 2425 (40%) of them had first sex with persons older by 5–10 years. The sexual practice was in hotels and bars in 2023 (33.2%) and was street-based in 1871 (30.7%). Alcohol dependence was report 2257 (37.2%) and chewing Chat in 3827 (62.4%) of respondents during the last thirty days before the survey.

Condom use was practiced in 5119 (84%) FSW. Of the remaining 966 (16%) who were not using condom, 23 (2.4%) were HBsAg positive, and among those who reported using condoms systematically at every sexual intercourse, 134 (2.7%) were HBsAg positive. HCVAb was positive in 4 and 23 of those who used condom consistently and inconsistently, respectively.

The first sex experience of the FSW was in 1347 (22.1%) forced, whereas in 4738 (77.9%) not forced. In the last six months preceding the study, 1645 (27%) of the FSW reported they had more than 90 clients. The majority, 5119 (84.1%), of the FSW utilized a condom during sex, and condom breakage was experienced by 1825 (30%).

## Prevalence of HBV and HCV co-infection with syphilis and HIV

Among the FSWs in the survey, 1768 (29.1) had a history of STI and 1140 were HIV-positive, of which 40 (3.5%) were coinfected with HBV, and 12 (1.1%) with HCV infection (Table 4). Syphilis was diagnosed in 339 (3.6%) of the FSW, and co-infection with HBV and HCV was 3.8% and 2.1%, respectively.

**Prevalence of HBV and HCV.** Of the 6085 FSW tested for HBV, 157 were tested positive for the HBsAg, a prevalence of 2.6% (95% CI: [2.2, 2.8]), and 27 FSW were positive for

**Table 4. Weighted prevalence of hepatitis B and hepatitis co-infection with syphilis and HIV among female sex workers, Ethiopia, 2020 (N = 6085).**

| Variables | | Frequency | Percent | 95% CI |
|---|---|---|---|---|
| History of STI | Syphilis | 339 | 5.6 | 5.0–6.2 |
| | Abnormal vaginal discharge | 887 | 14.6 | 12.8–15.6 |
| | Syphilis HIV co-infection | 161 | 2.6 | 2.3–3.1 |
| | Genital ulcer | 381 | 6.3 | 5.4–7.3 |
| | No history of STI | 4317 | 70.9 | 68.9–72.1 |
| HIV test result | Tested negative | 4945 | 81.3 | 80.3–82.2 |
| | Tested new positive | 565 | 9.3 | 8.6–10.0 |
| | Already known positive | 575 | 9.4 | 8.7–10.2 |
| Hepatitis B/HIV co-infection | HIV negative | 117 | 2.2 | 2.0–2.8 |
| | HIV positive | 40 | 3.5 | 2.6–4.7 |
| Hepatitis C/HIV co-infection | HIV negative | 15 | 0.3 | 0.2–0.5 |
| | HIV positive | 12 | 1.1 | 0.6–1.8 |
| Syphilis co-infection with HBV and HCV | HBV co-infection | 13 | 3.8 | 2.2–6.3 |
| | HCV co-infection | 7 | 2.1 | 0.9–4.0 |

**Table 5. Weighted prevalence of hepatitis B and hepatitis C infections among female sex workers, Ethiopia, 2020 (N = 6085).**

| Infections | Overall (N = 6085) | Percent (%) | 95% CI |
|---|---|---|---|
| HBV | 157 | 2.6 | 2.2–2.8 |
| HCV | 27 | 0.5 | 0.4–0.7 |
| HBV/HCV | 5 | 0.09 | 0.03–0.29 |
| HBV among non-condom users during sex (n = 966) | 23 | 2.4 | 2.0–3.4 |
| HBV among condom users during sex (n = 5119) | 134 | 2.6 | 2.2–2.8 |
| HCV among non-condom users during sex (n = 966) | 4 | 0.4 | 0.2–0.8 |
| HCV among condom users during sex (n = 5119) | 23 | 0.5 | 0.4–0.7 |

HCVAb, a prevalence of 0.5% (95% CI (0.4, 0.7). Only 5 (0.1%) of the participants were co-infected with both HBsAg and HCVAb. Among the 6085 participants in the study, a total of 184 were infected with HBV or had HCV Ab (Table 5).

**HBV and HCV prevalence in the study cities/towns.** As shown in Table 6, HBsAg prevalence was highest in Arba Minch (2.9%) followed by Addis Ababa (2.8%) and Adama city (2.3%). The highest percentage of HCVAb was most prevalent (2.9%) in Arba Minch, followed by Nekemite town (2.3%).

**Factors associated with hepatitis B and C.** Results of bivariate and multivariate logistic regression analyses are presented in Table 7. In the bivariate analysis, FSW with HBV infection had a significantly higher odds of being in the age groups 25–29 and 30–34, [COR = 1.2; 95% CI (1.01, 3.93), $P = 0.045$)] and [COR = 2.35; 95% CI (1.14, 4.81), $P = 0.02$)], respectively, compared with the age group 15–19 years. The odds of having over 90 sexual partners compared with those having under 30 partners in the past six months of being in age groups 20–24 or 25 years and above at first sex selling compared with those under 20, and of being HIV positive compared with being HIV negative was significant among FSW with HBV infection, [COR = 1.6; 95% CI (1.07, 2.4), $P = 0.072$)], [COR = 1.68; 95% CI (1.15, 2.46), $P = 0.008$)],

**Table 6. Weighted prevalence of hepatitis B and hepatitis C infections among female sex workers in the study cities/towns, Ethiopia, 2020 (N = 6085).**

| Residence (City/Town) | HBV (n = 157) | | HCV (n = 27) | |
|---|---|---|---|---|
| | Percent(%) | 95% CI | Percent(%) | 95% CI |
| Adama | 2.3 | (1.6 3.1) | 0.3 | (0.1, 0.7) |
| Addis Ababa | 2.8 | (2.3, 3.5) | 0.4 | (0.2, 0.7) |
| Arba Minch | 2.9 | (1.8, 4.3) | 2.9 | (1.8, 4.3) |
| Bahir Dar | 1.7 | (0.9, 3.4) | 1.3 | (0.6, 2.8) |
| Combolcha/Dessie | 6.0 | (3.7, 9.5) | 1.3 | (0.3, 3.0) |
| Dilla | 3.3 | (2.2, 4.8) | 0.0 | |
| Diredawa | 2.4 | (1.6,3.5) | 0.3 | (0.1, 0.8) |
| Gamebella | 1.9 | (0.9,3.4) | 0.0 | |
| Gonder | 3.8 | (2.4, 5.8) | 0.7 | (0.3, 1.9) |
| Harar | 1.4 | (0.7, 2.3) | 0.0 | |
| Hawassa | 2.7 | (1.7, 4.0) | 0.2 | (0.1, 0.9) |
| Jimma | 1.8 | (1.0, 2.8) | 0.0 | |
| Logia/Semera | 1.5 | (0.8, 2.6) | 0.9 | (0.4, 1.8) |
| Mizan | 3.0 | (2.0, 4.4) | 0.4 | (0.1, 1.1) |
| Nekemite | 2.3 | (0.9, 4.4) | 2.3 | (0.9, 4.4) |
| Shashemane | 1.7 | (0.9, 2.8) | 0.0 | |

**Table 7. Factors associated with hepatitis B and hepatitis C among female sex workers, Ethiopia, 2020.**

| Variable | | HBV (n = 157) | | | | HCV (n = 27) | | | |
|---|---|---|---|---|---|---|---|---|---|
| | | COR (95%C.I) | P-value | AOR (95% CI) | P-Value | COR (95%C.I) | P-value | AOR (95%CI) | P-value |
| Age | 15–19 | 1* | | | | | | | |
| | 20–24 | 1.15 (0.57, 2.33) | 0.390 | | | | | | |
| | 25–29 | 1.2 (1.01, 3.93)* | **0.045** | | | | | | |
| | 30–34 | 2.35 (1.14, 4.81)* | **0.02** | | | | | | |
| | 35–59 | 1.51 (0.7, 3.26) | 0.3 | | | | | | |
| Number of sexual partner in the past 6 months | < 30 | 1* | | 1* | | | | | |
| | 31–60 | 1.3 (0.84, 2.02) | 0.240 | 1.31 (0.84, 2.03) | 0.228 | | | | |
| | 61–90 | **1.6 (0.96, 2.69)*** | **0.072** | **1.66 (0.99, 2.79)**** | **0.054** | | | | |
| | 91+ | **1.6 (1.07, 2.40)*** | **0.021** | **1.66 (1.11, 2.49)**** | **0.013** | | | | |
| Average monthly income from selling sex in ETB | < 2500 | 1* | | | | | | | |
| | 2500–4999 | 1.16 (0.76, 1.77) | 0.496 | | | | | | |
| | 5000–7499 | 1.49 (0.94, 2.36) | 0.089 | | | | | | |
| | 7500+ | 1.37 (0.85, 2.22) | 0.198 | | | | | | |
| Moderate to severe depression | Not depressed | 1* | | | | 1* | | | |
| | Mild Depression | 0.72 (0.51, 1.02)* | 0.067 | | | 0.98 (0.39,2.47) | 0.961 | | |
| | Moderate to severe depression | 0.85 (0.55, 1.32)* | 0.474 | | | 2.27 (0.90–5.74) | 0.083 | | |
| Number of cities worked sex selling in the last three years | Same town | | | | | | | | |
| | 1 more town | 1.48 (0.97, 2.27) | 0.069 | | | | | | |
| | 2 or more towns | 1.48 (0.83, 2.66 | 0.186 | | | | | | |
| Number of non- paying partners in the past 6 month | Never | | | | | | | | |
| | Only one | 2.34 (0.86, 6.38 | 0.096 | | | | | | |
| | 2 and more | 1.98 (0.70, 5.64 | 0.198 | | | | | | |
| Age at first sex | 15 or less | | | | | 0.24 (0.08,0.70) | 0.009 | **0.21 (0.07,0.61)**** | **0.005** |
| | 16–20 | | | | | 0.17 (0.06,0.50) | 0.001 | **0.18 (0.061,0.53)**** | **0.002** |
| | 21+ | | | | | 1* | | | |
| Age at first sex selling | < 20 | 1* | | | | | | | |
| | 20–24 | **1.68 (1.15, 2.46)** | **0.008** | **1.67 (1.14, 2.44)**** | **0.009** | 0.59 (0.22,1.64) | 0.314 | | |
| | 25+ | **1.64 (1.06, 2.52)** | **0.025** | **1.56 (1.004, 2.43)**** | **0.048** | 1.83 (0.77,4.31) | 0.169 | | |

(*Continued*)

**Table 7.** (Continued)

| Variable | | HBV (n = 157) | | | | HCV (n = 27) | | | |
|---|---|---|---|---|---|---|---|---|---|
| | | COR (95%C.I) | P-value | AOR (95% CI) | P-Value | COR (95%C.I) | P-value | AOR (95%CI) | P-value |
| HIV Test Result | Tested Negative | 1* | | | | | | | |
| | Tested New Positive | 1.28 (0.76, 1.14) | 0.349 | 1.249 (0.74, 2.09) | 0.410 | 2.93 (1.06–8.10) | 0.038 | 2.05 (0.71,5.92) | 0.186 |
| | Known Positive | **1.72 (1.09, 2.71)*** | **0.020** | **1.64 (1.03, 2.62)**\*\* | **0.036** | **4.05 (1.64–9.98)*** | **0.002** | **2.85 (1.10,7.37)**\*\* | **0.031** |
| Syphilis | Non-Reactive | 1* | | | | | | | |
| | Reactive | 1.55 (0.87, 2.77) | 0.137 | | | **6.03 (2.53–14.38)** | **0.000** | **4.38 (1.73,11.11)** | **0.002** |

1*- reference (used as constant); *- significant on Bivariate analysis; **- significant at multivariate analysis

[COR = 1.64; 95%CI (1.06, 2.52), P =) .025)] and [COR = 1.72; 95% CI (1.09, 2.71), P = 0.020)] respectively. In the multivariate logistic regression analysis, having more than 90 sexual partners in the past six months compared with those having less than 30 partners, being in the age groups 20–24 and ≥25 at first sex selling compared with those under 20 years, and being HIV positive, [AOR = 1.66; 95% CI (1.11, 2.49), P = 0.013)], [AOR = 1.67; 95% CI (1.14, 2.44), P = 0.009)], [AOR = 1.56; 95% CI (1.004 2.43), P = 0.048)] and [AOR = 1.64; 95% CI (1.03, 2.62), P = 0.036)], respectively, are significantly and independently associated with HBV infection among FSW.

In the bivariate analysis, FSW with HCV infection had a lower odds of being in the younger aged group of 15 years or under was significantly higher [COR = 0.24, 95% CI (0.08,0.70), P = 0.009)] and 16–20 years [COR = 0.17; 95% CI (0.06,0.50), P = 0.001)] compared those aged above 20 years. The odds of being newly HIV positive or known HIV positive compared with being HIV negative and being positive for syphilis compared with being negative for syphilis was significant [COR = 2.93; 95% CI (1.06–8.10), P = 0.038)], [COR = 4.05; 95% CI (1.64–9.98); P = 0.002)] and [COR = 6.03; 95% CI (2.53, 14.38), P = 0.00)], respectively. In the multivariate logistic regression analysis, FSW with HCV infection is significantly and independently associated with age at first sex of 15 years or less [AOR = 0.21; 95% CI (0.07, 0.61), P = 0.005)] and age 16–20 years, compared with age of 25 years and above [AOR = 0.18; (0.061, 0.53), P = 0.002)] known HIV status [AOR = 2.85; 95% CI (1.10, 7.37), P = 0.031)] and being positive for syphilis [AOR = 4.38; 95% CI (1.73,11.11), P = 0.002)]

## Discussion

Among the 6085 FSW enrolled in this survey, the prevalence of HBV is 2.6% and of HCV 0.5%. According the WHO HBV infection prevalence classification−high (>8%), intermediate (2–8%), and low (<2%) [2] our finding falls in the intermediate category. The prevalence is lower than what has been reported from the rest of Africa and South-East Asia, but higher than that reported from the Americas and Eastern Mediterranean [2].

There is much variation in the prevalence HBV infection across countries, including those which have a lower prevalence than ours like Mexico 0.2% [25], Iran 1.1% [26], Greece 1.3% [27], Brazil 0.7% [28], those with similar prevalence to ours like Rwanda 2.5% [29], and countries in SSA having much higher prevalence than ours like Nigeria 17.1% [30], Kenya, 13.3% [31], and Ghana 15.0% [32]. Similarly, isolated and limited studies on HBV conducted in different cities/towns in Ethiopia at various times reported varying prevalence by site and year of

study—Dessie 13.1% [12], Hawassa 9.2% [19], Gonder 28.9 [20], and Mekelle 6% [21]. This variation could largely be explained by the differences in sociodemographic characteristics of study populations, study settings, sample size, and sampling methods focusing on high-risk population groups.

Our study showed that HBV prevalence was significantly associated with the age groups 25–29 years and 30–34 years, but this did not achieve significant independent association in the multivariable analysis. This is consistent with the finding by Forbi JC. et al [30] who reported a high HBV prevalence among 30–35 year-old FSW, but the difference by age group was not statistically significant. In contrast, Vázquez-Martínez, et al [33] reported that age above 30 years was significantly associated with HBV infection. It appears that FSW in the older age categories have an increased risk of getting HBV infection.

Having more than 90 sexual partners in the past six months was independently associated with HBV infection compared to the younger group in our survey. This association of HBV infection with having a greater number of clients indicates that FSW were at a higher risk of getting the infection because of an increase in exposure. Having a history of multiple sex partners increases the possibility of having sex with a partner during the acute phase of HBV infection. This has also been shown by previous studies in Ethiopia, India, Brazil, Egypt, Japan as well as by the WHO 2014 global network of sex work project on HIV and other STI in low and middle-income countries [19, 34–39].

Being in the age groups 20–24 and 25 years and above at first sex selling was significantly and independently associated with HBV infection compared with those under 20 years. This finding is inconsistent with study reports from Nigeria and Mexico [30, 33], which reported that early age of sexual activity would increase the risk of HBV infection. This difference might be an indication that FSW in our study had sexual involvement at an early stage and might have not received HBV vaccination in their early childhood because the national viral hepatitis prevention and control program in Ethiopia was initiated in 2016, whereas in Nigeria and Mexico, HBV vaccination is given in early childhood.

HIV positive status was significant associated with HBV infection among FSW in our survey. Studies showing such an association are scarce. However, a Rwandan study that looked at the HIV infection and syphilis co-infection among FSW indicated that HBV infection is an independent predictor of HIV and syphilis co-infection [29]. Use of condom is an efficient way of reducing the acquisition of STI, including HIV, syphilis and HBV infection. Our data showed that HBV infection was higher among FSW using condom, which could, as suggested by a report for Peru [40], be due to transmission through routes other than sexual intercourse. This is not supported by other reports [37, 41] which show inconsistent use of condoms is associated with increased risk HBV infection.

Our finding of HCV prevalence of 0.5% is similar to that reported form Nairobi, Kenya (0.76%) [31], and another study in Ethiopia (0.7%) [42]. It also concurs with the prevalence reported from the United States and Europe [43, 44] as well as the global average HCV prevalence of 0.8% [45]. In contrast, our finding is lower that reported from Iran of 6.2% [46], Ghana of 2.8% [47], and Port Harcourt, Nigeria 2.9% [48]. Similarly, syphilis seroprevalence of 5.6% from this study is higher than that of Ghana 7.5% [49], Tanzania 12.7% [50], and a previous report from Ethiopia 1.3% [42]. Our finding indicates that HCV prevalence was relatively lower than what has been reported by others. This difference might be related to the improvement in technology making current screening kits more specific and reliable, but could also be an indication of an improvement in national HCV prevention or a treatment programs. It could also be an indication of actual difference in HCV prevalence in different geographic settings.

FSW with HCV infection in our study who were over 20 years of age were at a greater risk for HCV infection. This concurs with the finding of a previous study, which showed low HCV

seroprevalence among younger FSW [51], and may be explained by the rise in HCVAb with age resulting from a continuous exposure to the virus.

Many HBV or HCV infected individuals were co-infected with HIV, mainly because of the nature of HIV route of transmission they share. FSW with HCV infection carried nearly a three-fold increase in the risk of being HIV positive in our study. This finding was supported with a study conducted in Burkina Faso West Africa [52], and the results of a systematic review in SSA [53]. HCV is usually spread through contact with blood of an infected person, which can happen while sharing drug injection equipment [54]. or rather commonly sexual among sexually active groups [55]. Nonetheless, the significant association between HCV and HIV infection among FSW has not been clearly described in published studies and this needs further investigation.

HCV positive status is associated with syphilis infection [56],which is also observed in our study. Moreover, a study conducted by Tessema B, et al. [42] suggested that the highest rate of co-infection and the statistically significant relationship between HCV and syphilis infections might be largely explained by the common modes of transmission and risk factors they share. This is in line with our finding that showed FSW with HCV infection were 4.4 times more likely to be syphilis sero-reactive. This suggests that prevention mechanisms and intervention need to be instituted among FSW to decrease further transmission of HCV and syphilis to the general population.

The previous prevalence estimates of HBV among the general population in Ethiopia ranged from 8%-12%, and HCV prevalence estimated at greater than 2.5% [16]. These findings were higher than our finding among FSW, a group at a much high risk of getting HBV/HCV and STI. It appears that the national estimates of these infections among the general population could have been overestimated. Overall, different studies conducted in Ethiopia on HBV and HCV have produced varying seroprevalence estimates. The studies have been conducted in different population groups carrying varying risks, utilized different sample sizes, and used different laboratory screening methods, some with and others without laboratory confirmatory testing. Moreover, the studies were conducted in different geographic settings. In light of this, large-scale seroprevalence and epidemiological studies may be required to ensure a more robust and current national seroprevalence HBV and HCV estimate.

The rates of HBV and HCV co-infection among HIV positive FSW in the present study were similar to what has been reported from the Global Prevalence of HBsAg and HIV and HCV Ab study [57]. Moreover, the rate of acute HBV infection will have major implication to the epidemiology of chronic HBV infection as reported from in different settings [58–61]. In Ethiopia, which is a low-income country, HBV transmission is like to be through drug injection and high-risk sexual behaviors [62], and the prevalence identified in this study puts it in the moderate risk endemicity category. This is much lower than the level of endemicity in the reset of Africa, except Tunisia and Morocco [57, 63], which have an endemicity level similar to Ethiopia. According to WHO, 80% of people with viral hepatitis live without access to prevention, testing, and treatment services [64], and this requires due attention particularly in developing setting where HBV vaccine coverage is very low.

## Strengths and limitations

The main strength of our study was the inclusion of participants from the high-risk groups and that the study this is the first report on the prevalence of and HBV and HCV among FSW in Ethiopia involving all regional capital cities and selected major towns of the country. The survey targeted only regional capitals, and major towns of the country. The presence of harder-to-reach FSW like home-based sex workers might not be fully accounted for, and lack of systematic naming of the streets in the survey cities and towns was a challenge in the

mapping of locations. Enzyme immunoassay could have helped in to confirm the results was very helpful, but was not done because of constraints with budget, and it is likely that there were some false-positive results. These factors need to be taken into consideration in the interpretation of the survey results.

## Conclusion

This study reveals an intermediate prevalence of HBV and a low prevalence of HCV infections among FSW in regional capitals and selected major town of Ethiopia. The prevalence we identified is lower than the estimate among the general population reported by previous studies. While this might have been influenced by difference in methodologies, our finding may also suggest that the coverage of hepatitis related interventions among FSW has been effective, but require strengthening to control HBV and HCV transmission among FSW. Scaling-up interventions for FSW such as HBV vaccination, reducing number of sexual partners, increasing condom distribution and regularly monitoring and screening for these infections and other STI among FSW is needed. FSW who are HBV and HCV infected should be informed about the transmission routes and methods to prevent further spread of the virus. FSW testing HBV negative but not yet got vaccinated need to receive the vaccine since they were at high-risk of contracting the infection. Further epidemiological studies to determine the prevalence and determinants of HBV, HCV and other STI among different population groups are suggested.

## Acknowledgments

The authors would like to acknowledge the Ethiopian Public Health Institute for providing materials support during this project implementation. We also thank Ethiopian Public Health institute, National HIV/AIDS surveillance and laboratory treatment center staff members for their cooperation during data collection. We would also like to acknowledge all FSW and their partners whose data were used in this study, and all the healthcare workers who took part in the educating and treating of the included FSW.

## Author Contributions

**Conceptualization:** Birra Bejiga Bedassa.

**Data curation:** Birra Bejiga Bedassa, Gemechu Gudeta Ebo, Jemal Ayalew Yimam, Feyiso Bati Wariso, Sileshi Lulseged, Saro Abdella Abrahim.

**Formal analysis:** Birra Bejiga Bedassa, Gemechu Gudeta Ebo, Jemal Ayalew Yimam.

**Funding acquisition:** Getachew Tollera Eticha, Tsigereda Kifle Wolde, Saro Abdella Abrahim.

**Investigation:** Birra Bejiga Bedassa.

**Methodology:** Birra Bejiga Bedassa.

**Project administration:** Jaleta Bulti Tura, Getachew Tollera Eticha, Tsigereda Kifle Wolde, Saro Abdella Abrahim.

**Resources:** Jaleta Bulti Tura, Saro Abdella Abrahim.

**Software:** Gemechu Gudeta Ebo, Jemal Ayalew Yimam, Feyiso Bati Wariso.

**Supervision:** Birra Bejiga Bedassa, Gemechu Gudeta Ebo, Jemal Ayalew Yimam, Jaleta Bulti Tura.

**Validation:** Birra Bejiga Bedassa, Sileshi Lulseged.

**Visualization:** Birra Bejiga Bedassa, Sileshi Lulseged.

**Writing – original draft:** Birra Bejiga Bedassa, Gemechu Gudeta Ebo, Jemal Ayalew Yimam, Feyiso Bati Wariso.

**Writing – review & editing:** Gemechu Gudeta Ebo, Jemal Ayalew Yimam, Sileshi Lulseged, Saro Abdella Abrahim.

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
