## [Decision Letter · Decision Letter 0]

8 Aug 2022

PONE-D-22-14713Prevalence and factors associated with hepatitis B and C virus infections among female Sex workers in Ethiopia: Results of the national biobehavioral Survey, 2020PLOS ONE

Dear Dr. Bejiga,

Thank you for submitting your manuscript to PLOS ONE. After careful consideration, we feel that it has merit but does not fully meet PLOS ONE’s publication criteria as it currently stands. Therefore, we invite you to submit a revised version of the manuscript that addresses the points raised during the review process.

 Your manuscript was reviewed by 3 experts in the field. They identified many important problems in your submission and provided helpful comments. Please review these comments and provide point-by-point responses.==============================

We look forward to receiving your revised manuscript.

Kind regards,

Yury E Khudyakov, PhD

Academic Editor

PLOS ONE

Journal Requirements:

“This project was conducted using the effort of many institutions, organizations, and individuals without whose contributions could not have been possible like, FHAPCO, MOH, CDC, ICAP, PSI and Regional health bureau the U.S. President’s Emergency Plan for AIDS Relief (PEPFAR) funds obtained though the U.S Center for Disease Control and Prevention (CDC) under the term of cooperative agreement #U2GGH001226. The findings and conclusions in this report are those of the authors and do not necessarily represent the official position of the funding agency.”

Reviewers' comments:

Reviewer's Responses to Questions

**Comments to the Author**

1. Is the manuscript technically sound, and do the data support the conclusions?

Reviewer #1: Yes

Reviewer #2: Yes

Reviewer #3: Yes

2. Has the statistical analysis been performed appropriately and rigorously? 

Reviewer #1: No

Reviewer #2: Yes

Reviewer #3: Yes

3. Have the authors made all data underlying the findings in their manuscript fully available?

Reviewer #1: Yes

Reviewer #2: Yes

Reviewer #3: Yes

4. Is the manuscript presented in an intelligible fashion and written in standard English?

Reviewer #1: No

Reviewer #2: Yes

Reviewer #3: Yes

5. Review Comments to the Author

Reviewer #1: The title of the "Prevalence and factors associated with hepatitis B and C virus infections among female Sex workers in Ethiopia: Results of the national biobehavioral Survey, 2020" topic is interesting.

The sample size is big enough to provide reliable results.

However, authors must clarify the highlighted comments in the attached document.

The literature review is poor in the introduction section.

There is inconsistency in the method section.

The information of Ethiopia background should be summarized and focus on settings where the study have been conducted.

How towns and cities were selected?

There are no FSW in rural settings?

It is not national representative study because some areas were excluded

- Authors should specify the name of tests used in the study.

- Test algorithm

- Sample storage and transportation.

- Clinical management once the test is positive

- Syphilis and HIV results are not part of the study.

In the abstract section authors mentioned that <=0.05 p-value was considered. 0.2 p-value is too high, it can be considered in some few key variables

Median=25 and IQR=8?

Check please

1. Consistency in data analysis:

- Age group categories with p-values <0.05 not appearing in multivariable model. The same for number of sexual partners

2. Multivariable model without bivariate model. What happened? Age of first sex

3. Age at first sex selling not completed

The discussion section is not structured.

The author is reporting the results

The reader will be bored with this so long and light discussion

Reviewer #2: The manuscript by Bedassa et al. is well written and provides important information on the prevalence of HBV and HCV infection in Ethiopian FSWs. Although the main objective of the study was HBV and HCV, the authors provide important information about the prevalence of HIV and syphilis in this population. In my opinion, the manuscript should be published after minor modifications.

1. Authors need to add study exclusion criteria. Only the inclusion criteria were placed.

2. Which kit manufacturers are used?

3. It would be interesting to include the index of FSWs who did not accept to participate in the study.

4. An important detail is the higher prevalence of HBV and anti-HCV in FSWs who reported using codons in their sexual intercourse. What could justify this finding?

5. The prevalence of HBV and anti-HCV in the study was much lower than that observed in the general population of Ethiopia. The authors suggest that this can be influenced by different methodologies. Wouldn't it be interesting to carry out an enzyme immunoassay to confirm the results (if there were financial resources for this)? Could they have too many false negatives? This is a very interesting result of the study. Another important point would be to verify the vaccination coverage for hepatitis B in this population.

Reviewer #3: The authors did a great job giving relevance to aspects of the epidemiology of infectious agents in vulnerable populations and a very rich statistical analysis with a huge sample.

Nonetheless, some excerpts can be improved to provide a better reading and understanding of the article.

1. The authors should clarify the following sections to avoid confusion - lines 254 trhough 259 - the excerpt should focus on co-infection with HIV-positive cases despite HIV-negative cases.

2. Tables 1 to 4 should include a column to add the 95% CI

3. Table 1 presents duplicate entries

4. There are some misspelled words - line 65 "HVC"; line 115-116 duplicated "in 2016"; line 122 - "crass"; line 184 "a liquated"; line 239 "chat"; line 354 "STS".

5. In the results section, under the topic Factors associated with hepatitis B and C, please provide a clearer explanation of what the COR parameter is about. Is it the same AOR presented in the data analysis topic and was it just typed wrong? if so, change the acronym COR to AOR whenever it's mistaken, if not, provide an adequate explanation of how this statistical result was obtained and its purpose.

6. At the beginning of the discussion section, lines 355 and 356, it would be better to present the reference right after the mention of the previous study cited in this paragraph.

7. Review the score on line 321, there should be a point after endemicity.

8. The authors should revise the language to improve readability in lines 340 to 344 to make it more understandable. Do the same for the lines 357 and 358; 371 to 374; 385 to 389; 395 to 397,on line 436 where it starts with ... , a group.

6. PLOS authors have the option to publish the peer review history of their article (what does this mean?). If published, this will include your full peer review and any attached files.

Reviewer #1: **Yes: **Mutagoma Mwumvaneza

Reviewer #2: **Yes: **Luiz Fernando Almeida Machado

Reviewer #3: **Yes: **Rogério Valois Laurentino

---

## [Author Response · Author response to Decision Letter 0]

15 Oct 2022

We deeply appreciate the editor’s and reviewers' comments and suggestion, which found very helpful to in improving on the quality of the manuscript.

---

## [Decision Letter · Decision Letter 1]

31 Oct 2022

PONE-D-22-14713R1Prevalence and factors associated with hepatitis B and C virus infections among female Sex workers in Ethiopia: Results of the national biobehavioral Survey, 2020.PLOS ONE

Dear Dr. Bejiga,

Thank you for submitting your manuscript to PLOS ONE. After careful consideration, we feel that it has merit but does not fully meet PLOS ONE’s publication criteria as it currently stands. Therefore, we invite you to submit a revised version of the manuscript that addresses the points raised during the review process.

Your revised manuscript was reviewed by 2 experts in the field. Both identified some remaining problems in the manuscript which require your careful attention. Please consider the attached comments and provide point-by-point responses.

We look forward to receiving your revised manuscript.

Kind regards,

Yury E Khudyakov, PhD

Academic Editor

PLOS ONE

Journal Requirements:

Reviewers' comments:

Reviewer's Responses to Questions

**Comments to the Author**

1. If the authors have adequately addressed your comments raised in a previous round of review and you feel that this manuscript is now acceptable for publication, you may indicate that here to bypass the “Comments to the Author” section, enter your conflict of interest statement in the “Confidential to Editor” section, and submit your "Accept" recommendation.

Reviewer #2: (No Response)

Reviewer #3: (No Response)

2. Is the manuscript technically sound, and do the data support the conclusions?

Reviewer #2: Yes

Reviewer #3: Yes

3. Has the statistical analysis been performed appropriately and rigorously? 

Reviewer #2: Yes

Reviewer #3: Yes

4. Have the authors made all data underlying the findings in their manuscript fully available?

Reviewer #2: Yes

Reviewer #3: Yes

5. Is the manuscript presented in an intelligible fashion and written in standard English?

Reviewer #2: Yes

Reviewer #3: Yes

6. Review Comments to the Author

Reviewer #2: I would like to thank you for the opportunity to review the manuscript by Bedassa et al. and congratulate the authors for the information regarding the epidemiology of HBV and HCV in Ethiopian FSWs. The manuscript is well written but, in my opinion, needs some adjustments to be published.

Here are my observations:

- Line 66: put HIV in full before the acronym

- In the introduction put the prevalence of HCV in FSWs in Ethiopia

- Line 295: state that 184 were infected with HBV or had HCV Ab, because from the way it is written, you can understand that it was HBV-HCV co-infection.

- My main suggestion for authors to add the prevalence of HBV and HCV Ab in each city (could be in a table) to demonstrate the location(s) of highest prevalence, which could help in decision making regarding the fight against infection by the health authorities.

Reviewer #3: There are several typing corrections that should be looked at more carefully considering maintaining a standard in writing the article, this attention must be given to the spacing of the text for the references that are in parentheses, for example, all citations should follow the same pattern presented in line 63, being clearer and more specific, several times the space was not given in typing between the text and parenthesis of the reference, this needs to be corrected in lines 65, 70, 72, 74, 88, 89, 90, 93, 95, 98, 103, 105, 222, 339, 341, 342, 343, 344, 346, 382, 385, 387, 394, 399, 402, 413. In all these lines, the typing should be like this: are endemic in developing countries (1) [note the space before the quote] instead of like this: .. ..mostly from hepatitis-related cirrhosis and liver cancer(2) [without the space before the citation].

In line 103, a space must be inserted after the period in the excerpt: ...(12).FSW have been identified as a population group with the highest risk for STI...

In lines 106/107 it should be written: to determine instead of to determined in the excerpt: there is no national data among FSW to determined HBV and HCV prevalence and driving factors

In line 114 the word "in" after the period must be capitalized in the passage: ... 120 million and had low per capita income (22). in 2016, when the...

In line 188 it should be written "aliquoted" instead of "a liquidated" in the passage: the plasma was separated and a liquidated in two...

In line 192, a space must be inserted before the abbreviation EPHI in the passage: teams from Ethiopian Public Health institute(EPHI).

In table 1, the font used in the 95% confidence interval column is clearly one size larger than the rest of the table, standardizing for the same font and font size

In line 363 the dot must be after the second parenthesis in the passage: countries.(18, 36-41)

In line 373 it should be "scarce" instead of "scare"

In line 388 it should be "might be related to" instead of "might related to"

In line 400 it should be "through" instead of "though"

Line 401 should not have the dot in the section: ...injection equipment (56).or rather commonly...

In line 443 it should be: "These factors need to be taken" instead of "These need factors to be taken into"

7. PLOS authors have the option to publish the peer review history of their article (what does this mean?). If published, this will include your full peer review and any attached files.

Reviewer #2: No

Reviewer #3: **Yes: **Rogério Valois Laurentino

---

## [Author Response · Author response to Decision Letter 1]

22 Nov 2022

NOV 22, 2022

PONE-D-22-14713

Manuscript Title: Prevalence and factors associated with hepatitis B and C virus infections among female Sex workers in Ethiopia: Results of the national biobehavioral Survey, 2020

Dear Editor,

On behalf of myself and the authors, I would like to thank you for a and the reviewers for having critically reviewed the manuscript and for giving allowing us to revise and re-resubmit it to PLOS ONE. We have revised the manuscript based on the comments and suggestions from you and all two reviewers. 

We deeply appreciate yours and the reviewers' comments and suggestions, which we found very helpful and used to improve the quality of the manuscript. Per your advice, we have prepared a point-by-point response to all comments as presented below. We have also made changes as advised in a separate file titled 'Revised Manuscript with Track Changes' and have uploaded the file per PLOS guidelines. In addition, we have uploaded three documents as part of our the online submission: the tracked and clean copies of the manuscript as well as a rebuttal letter. 

Sincerely

Birra Bejiga 

Journal Requirements:

Please review your reference list to ensure that it is complete and correct. If you have cited papers that have been retracted, please include the rationale for doing so in the manuscript text, or remove these references and replace them with relevant current references. Any changes to the reference list should be mentioned in the rebuttal letter that accompanies your revised manuscript. If you need to cite a retracted article, indicate the article’s retracted status in the References list and also include a citation and full reference for the retraction notice

Authors’ response: We have double-checked and ensured that all is complete and correct.

Reviewer's Responses to Questions

Comments to the Author 

1. If the authors have adequately addressed your comments raised in a previous round of review and you feel that this manuscript is now acceptable for publication, you may indicate that here to bypass the “Comments to the Author” section, enter your conflict of interest statement in the “Confidential to Editor” section, and submit your "Accept" recommendation.

Reviewer #2: (No Response)

Reviewer #3: (No Response)

2. Is the manuscript technically sound, and do the data support the conclusions?

Reviewer #2: Yes

Reviewer #3: Yes

Authors’ response: Thanks. Noted.

3. Has the statistical analysis been performed appropriately and rigorously?

Reviewer #2: Yes

Reviewer #3: Yes

Authors’ response: Thanks. Noted.

4. Have the authors made all data underlying the findings in their manuscript fully available?

Reviewer #2: Yes

Reviewer #3: Yes

Authors’ response: Thanks. Noted.

5. Is the manuscript presented in an intelligible fashion and written in standard English?

Reviewer #2: Yes

Reviewer #3: Yes

Authors’ response: Thanks. Noted.

6. Review Comments to the Author

Reviewer #2: I would like to thank you for the opportunity to review the manuscript by Bedassa et al. and congratulate the authors for the information regarding the epidemiology of HBV and HCV in Ethiopian FSWs. The manuscript is well written but, in my opinion, needs some adjustments to be published. 

Authors’ response: Thanks. Noted.

 Line 66: put HIV in full before the acronym

Authors’ response: lines 66 revised as suggested

In the introduction put the prevalence of HCV in FSWs in Ethiopia

Authors’ response: revised and added as suggested.

- Line 295: state that 184 were infected with HBV or had HCV Ab, because from the way it is written, you can understand that it was HBV-HCV co-infection.

 Authors’ response: lines 295 revised as suggested.

My main suggestion for authors to add the prevalence of HBV and HCV Ab in each city (could be in a table) to demonstrate the location(s) of highest prevalence, which could help in decision making regarding the fight against infection by the health authorities. 

 Authors’ response: revised and added as suggested.

Reviewer #3: There are several typing corrections that should be looked at more carefully considering maintaining a standard in writing the article, this attention must be given to the spacing of the text for the references that are in parentheses, for example, all citations should follow the same pattern presented in line 63, being clearer and more specific, several times the space was not given in typing between the text and parenthesis of the reference, this needs to be corrected in lines 65, 70, 72, 74, 88, 89, 90, 93, 95, 98, 103, 105, 222, 339, 341, 342, 343, 344, 346, 382, 385, 387, 394, 399, 402, 413. In all these lines, the typing should be like this: are endemic in developing countries (1) [note the space before the quote] instead of like this: .. ..mostly from hepatitis-related cirrhosis and liver cancer(2) [without the space before the citation].

 Authors’ response: thank you Revised as suggested. 

In line 103, a space must be inserted after the period in the excerpt: ...(12).FSW have been identified as a population group with the highest risk for STI...

 Authors’ response: thank you Revised as suggested. 

In lines 106/107 it should be written: to determine instead of to determined in the excerpt: there is no national data among FSW to determined HBV and HCV prevalence and driving factors

 Authors’ response: thank you for your observation Revised as suggested. 

In line 114 the word "in" after the period must be capitalized in the passage: ... 120 million and had low per capita income (22). in 2016, when the...

 Authors’ response: thank you Revised as suggested. 

In line 188 it should be written "aliquoted" instead of "a liquidated" in the passage: the plasma was separated and a liquidated in two...

 Authors’ response: thank you Revised

In line 192, a space must be inserted before the abbreviation EPHI in the passage: teams from Ethiopian Public Health institute(EPHI).

 Authors’ response: thank you Revised

In table 1, the font used in the 95% confidence interval column is clearly one size larger than the rest of the table, standardizing for the same font and font size

 Authors’ response: thank you Revised as suggested. 

In line 363 the dot must be after the second parenthesis in the passage: countries.(18, 36-41)

 Authors’ response: thank you Revised as suggested. 

In line 373 it should be "scarce" instead of "scare"

 Authors’ response: thank you Revised as suggested. 

In line 388 it should be "might be related to" instead of "might related to"

 Authors’ response: thank you Revised

In line 400 it should be "through" instead of "though"

 Authors’ response: thank you Revised as suggested. 

Line 401 should not have the dot in the section: ...injection equipment (56).or rather commonly...

 Authors’ response: thank you Revised

In line 443 it should be: "These factors need to be taken" instead of "These need factors to be taken into"

 Authors’ response: thank you Revised

7. PLOS authors have the option to publish the peer review history of their article (what does this mean?). If published, this will include your full peer review and any attached files.

Do you want your identity to be public for this peer review? For information about this choice, including consent withdrawal, please see our Privacy Policy.

Reviewer #2: No

Reviewer #3: Yes: Rogério Valois Laurentino

In compliance with data protection regulations, you may request that we remove your personal registration details at any time. (Remove my information/details). Please contact the publication office if you have any questions.-- 

With Best Regards

 Birra Bejiga Bedassa (MPH, Epidemiology and Biostatistics) 

Ethiopian Public Health Institute TB/HIV Directorate; HIV Surveillance Team

 Email: birr4allephi@gmail.com/ birr4all@gmail.com

ORcid:https://orcid.org/my-orcid?orcid=0000-0002-2784-3564

contact Number: +251 913687219

---

## [Editor Report · Decision Letter 2]

24 Nov 2022

Prevalence and factors associated with hepatitis B and C virus infections among female Sex workers in Ethiopia: Results of the national biobehavioral Survey, 2020.

PONE-D-22-14713R2

Dear Dr. Bejiga,

We’re pleased to inform you that your manuscript has been judged scientifically suitable for publication and will be formally accepted for publication once it meets all outstanding technical requirements.

Kind regards,

Yury E Khudyakov, PhD

Academic Editor

PLOS ONE
---

## [Editor Report · Acceptance letter]

19 Dec 2022

PONE-D-22-14713R2 

Prevalence and factors associated with hepatitis B and C virus infections among female Sex workers in Ethiopia: Results of the national biobehavioral Survey, 2020 

Dear Dr. Bedassa:

I'm pleased to inform you that your manuscript has been deemed suitable for publication in PLOS ONE. Congratulations! Your manuscript is now with our production department. 

Kind regards, 

on behalf of

Dr. Yury E Khudyakov 

Academic Editor

PLOS ONE